# Preparation and Performance of Multilayer Si-B-C-N/Diamond-like Carbon Gradient Films

**DOI:** 10.3390/ma16041665

**Published:** 2023-02-16

**Authors:** Jiaqi Duan, Minghan Li, Wenzhi Wang, Ziming Huang, Hong Jiang, Yanping Ma

**Affiliations:** 1Special Glass Key Lab of Hainan Province & State Key Laboratory of Marine Resource Utilization in South China Sea, Hainan University, Haikou 570228, China; 2State Key Laboratory of Special Glass, Chengmai, Haikou 570228, China; 3Hainan Haikong Special Glass Technology Co., Ltd., Chengmai, Haikou 570228, China

**Keywords:** diamond-like carbon films (DLC), gradient film, adhesive force, tribological performance

## Abstract

Si-B-C-N/diamond-like carbon (DLC) gradient films with different layers were prepared on a glass substrate by radio frequency magnetron sputtering, and the structure and surface morphology of the resulting films were analyzed by scanning electron microscopy, Raman spectrometry, and X-ray photoelectron spectroscopy. The mechanical and optical properties of the films were studied using a multifunctional material mechanical testing system, UV-Vis spectrophotometer, and micro-Vickers hardness tester. The gradient structure promotes the formation of sp^3^ bonds and improves the hardness and optical transmittance of the resulting films. Among the prepared films, the single-layer Si-B-C-N/DLC gradient film shows the highest optical transmittance (97%). Film–substrate adherence is strengthened by the introduction of the gradient structure. The best adhesion was obtained with a double-layer Si-B-C-N/DLC gradient film. Suitable anti-wear properties were exhibited in both dry (0.18) and wet (0.07) conditions. In this paper, evaluation of the microstructural, optical, and mechanical properties of the films could provide new insights into improvements in the bonding force of glass-based DLC films and enrich the experimental data of DLC multilayer film systems.

## 1. Introduction

Glass, a material that can be found everywhere in daily life, is widely used in optical windows, such as cell phone screens and watch dials, owing to its exceptional visible light transmission. However, during its use, the surface of glass develops scratches, which can seriously affect its quality and service life. Thus, improving the scratch resistance and anti-wear performance of glass is of great importance to expand its practical applications. Diamond-like carbon (DLC) films are a class of sub-stable amorphous carbon materials consisting of a mixture of diamond (with heterogeneous sp^3^ bonds) and graphite (with heterogeneous sp^2^ bonds) structures [1]. These films have desirable properties, such as high hardness, high wear resistance, low friction coefficient, and adequate chemical inertness and optical transmittance; thus, they have high application value as protective coatings on material surfaces [2,3,4]. However, DLC films cannot easily form chemical bonds owing to their excessive internal stress and low matching with the glass lattice. Moreover, DLC films deposited on glass are highly susceptible to cracking and peeling. In efforts to solve such problems, researchers have found that careful design of multilayer film structures can effectively address this issue via the uniform insertion of intermediate layers, which can effectively reduce the stress concentration and crack extension, thereby rendering the film stiff and elastic while releasing internal stresses [5]. Yang et al. [6] prepared DLC/CNx multilayer films on Si substrates. Studies have shown that the hardness of multilayer films can reach 21 GPa, the bonding force of the film base is 54 N, and the friction factor in the atmosphere is about 0.19, and found that the design of the multilayer film structure not only improved the hardness of the films but also reduced their internal stress and friction factor. Xu et al. [7] deposited DLC multilayers consisting of soft and hard DLC layers on Ti64 alloy substrates and found that as the modulation ratio decreases, the sp^3^ content and hardness of multilayer DLC films tend to decrease. The multilayer DLC films with a modulation ratio of 1:1 have the best wear resistance due to the balance between hardness and residual stress. To further reduce the internal stresses in the films, researchers have found that a gradient film preparation method can be used. Gradient multilayer films are mainly used to reduce film stresses by reducing the differences arising from elemental mutations between film layers (which results in high stress). Gradient multilayer films, as opposed to ordinary multilayer films, can efficiently reduce interfacial effects and layer mismatch issues, improving the bonding of the film substrate. Sheeja et al. [8] deposited thick, soft layers under a high pulse bias of several kilovolts before plating the harder films so that the stresses in the hard layers were relieved in the soft layers. Such a high pulse bias also caused a mixed layer to form between the hard substrate and the soft layers, and a gradient change occurred in its composition, thus improving the adhesion of the film layer to a greater extent. Some researchers [9] have also utilized a series of transition layers with gradients in composition, such as Ti, TiN, TiC_X_N_Y_, and TiC, and finally deposited DLC films, and the resulting internal stresses were greatly reduced.

Si-B-C-N ceramics are typical precursor ceramics with adequate thermal stability, high-temperature mechanical properties, and high-temperature creep resistance [10]. Si-B-C-N films prepared by magnetron sputtering have smooth and flat surfaces, form amorphous structures, and exhibit adequate structural stability and thermal conductivity and thermal expansion coefficients [11,12]. Many studies on Si-B-C-N thin films have been reported. J. Vlček et al. [13], for example, deposited amorphous Si-B-C-N films with smooth surfaces on SiC and Cu floating substrates and showed that the films had an elastic recovery of 75% with suitable optical transmittance and oxidation resistance. Onoprienko A A et al. [14] prepared Si-B-C-N films with controlled N contents and high hardness by DC reactive magnetron sputtering; the authors’ results [15] revealed the potential of B-C-N synergy in the design of new materials with enhanced properties.

In view of the performance enhancement brought about by multilayer systems and the potential of Si-B-C-N films, nano multilayers prepared from DLC and Si-B-C-N may be expected to yield films with exceptional overall performance despite the inherent interfacial effects of multilayer structures, which limit the bonding strength between layers. Therefore, in this work, Si-B-C-N/DLC gradient films with an anisotropic distribution of Si, B, N, and C are used instead of conventional Si-B-C-N/DLC layers with fixed contents, and multilayer Si-B-C-N/DLC gradient films are designed. Evaluation of the microstructural, optical, and mechanical properties of the films could provide new insights into improvements in the bonding force of glass-based DLC films and enrich the experimental data of DLC multilayer film systems.

## 2. Experimental Section

### 2.1. Sample Preparation

Si-B-C-N/DLC gradient films were deposited onto glass substrates (25.4 × 76.2 × 1.2 mm^3^, maximum visible light transmittance, 92%; Tonglihang Chemical Materials, Haikou, China) using a PVD 500 high-vacuum magnetron sputtering coating system (Sky Technology Development, Shenyang, China) with an AE radio frequency power source (13.56 MHz), argon (purity: 99.995%; Jin Hou Special Gas, China) as the working gas, and graphite (purity: 99.999%; Zhongnuo Advanced Material, Beijing, China) and Si-B-C-N (atomic ratio: 2:1:2.5:1; Zhongnuo Advanced Material, Beijing, China) as the sputtering targets. Figure 1 shows the deposition system geometry. Before coating, the glass substrate was ultrasonically cleaned with detergent, acetone, alcohol, and ultrapure water and then blown dry with N_2_. The substrate was then placed inside the vacuum chamber, and thin-film samples were prepared using the process parameters listed in Table 1. Samples 1 and 2, which were composed of pure DLC films and DLC films with a Si-B-C-N transition layer, respectively, were designated as control groups. Samples 3, 4, 5, and 6 were composed of single-, double-, triple-, and quadruple-layer Si-B-C-N/DLC gradient films, respectively. The sputtering power was linearly increased and decreased in the range of 60–130 W. The sputtering power change rates of the two targets for the single-, double-, triple-, and quadruple-layer gradient films were 0.77, 1.55, 2.3, and 3.1 W/min, respectively. During the sputtering process, the substrate temperature in the sputtering chamber was maintained at 60 °C, the argon flow rate was 7 × 10^−7^ m^3^/s, and the working air pressure was 1.5 Pa (determined by an orthogonal experiment through optical transmittance, I_D_/I_G_, and hardness analyses). The target–substrate distance was 150 mm, the baseline pressure before deposition was 2.0 × 10^−3^ Pa, and the total sputtering time was fixed at 90 min.

### 2.2. Sample Characterization and Performance Testing

The bond structures of the Si-B-C-N/DLC thin films were analyzed using an inVia Reflex laser Raman spectrometer over the wavenumber range of 800–2000 cm^−1^ with a laser wavelength of 514 nm and an exposure time of 10 s. The surface morphology and cross-sectional thickness of the films were analyzed using a Verios G4 UC Thermo Scientific field emission scanning electron microscope (SEM). The transmittance of the films between 300 and 1100 nm was tested using a Lambda 750S UV-Vis spectrophotometer with a step size of 1 nm. The composition and chemical bond structures of the films were analyzed using an Axis Supra (DAOJINKRATOS) X-ray photoelectron spectrometer (anode target: Al Kα 1486.6 eV, 0.68 eV/C1s). Prior to analysis, the surface of the films was etched by Ar ion bombardment to remove surface adsorbates. The friction coefficient and bonding force of the films were analyzed using an APEX UMT series multifunctional material mechanical testing system. Here, the friction piece was a Ø 1.6 mm tungsten carbide ball (Bruker Corporation, Karlsruhe, Germany), the friction path was a Ø 9 mm circle, the disc speed was 100 rpm, the loading time was held constant at 200 mN, and the loading time was 30 min. A micro-tip needle with a diameter of 5 μm was used to scratch the film surface, and the load was gradually and uniformly increased to 200 mN over 200 s.

## 3. Results and Discussion

### 3.1. XPS Analysis

As shown in Table 1, the outermost layer of each Si-B-C-N/DLC gradient film was prepared using only two cases. One was prepared with sputtering powers of 60 W for the C target and 130 W for the Si-B-C-N targets (e.g., Samples 4 and 6), and the other was prepared with sputtering powers of 130 W for the C target and 60 W for the Si-B-C-N targets (e.g., Samples 3 and 5). Thus, Samples 4 and 5 were selected as representative specimens for XPS analysis. The final characteristic peaks and split peak fitting of Si 2p, B 1s, C 1s, and N 1s are shown in Figure 2. Figure 2a shows the XPS surface full spectra of the two samples. Peaks with binding energies of 101.8, 102.8, and 103.0 eV in the XPS profile of Si 2p (Figure 2b) correspond to Si-N, Si-C, and Si-O bonds, respectively [16,17]. Peaks with binding energies of 191.3, 192.5, and 193.5 eV in the XPS profile of B 1s (Figure 2c) correspond to B-N, B-O, and B-C-N bonds, respectively [18]. Peaks with binding energies of 284.2, 284.8, 285.3, 286.6, and 288.4 eV in the XPS profile of C 1s (Figure 2d) correspond to C-B, C=C, C-C/C-Si, C-N, and C-O bonds, respectively [16]. Finally, peaks with binding energies of 398.4, 399.2, and 400.02 eV in the XPS profile of N 1s (Figure 2e) correspond to N-B, N-C, and N-Si bonds, respectively [19,20].

The relative content of each element in the films can be determined by calculating the area percentage occupied by each peak, the results of which are shown in Figure 2f. When the C target is bombarded at a sputtering power of 130 W and the Si-B-C-N target is bombarded at a sputtering power of 60 W, the surface layer of Sample 5 corresponds to a DLC film with C (C-B, C-C, C-Si, C-N) contents of up to 82%, Si (Si-N, Si-C, Si-B) content of 8%, and B and N (B-C-N, B-Si, N-C, N-Si) contents of 5%. When the C target is bombarded at a sputtering power of 60 W and the Si-B-C-N target is bombarded at a sputtering power of 130 W, the surface layer of Sample 4 corresponds to a Si-B-C-N/DLC composite film with C content of 40%, Si content of 37%, and B and N contents of 18%. It is also obvious that the elemental content of the surface layer of the two samples shows a large difference by each bonding bond, mainly due to the different sputtering ability of the two targets. High sputtering powers increase the size of particles bombarded from the target surface and accelerate the deposition rate [21]. Thus, the particle size of the clusters formed by one target on the limited glass substrate gradually increases and the effective deposition of the other target decreases.

### 3.2. Raman Spectral Analysis

Figure 3a shows the Raman spectra of Samples 2–6. The films show asymmetric broad peaks in the range of 1000–1800 cm^−1^, which are decomposed into D peaks near 1350 cm^−1^ by the transverse vibration of the sp^2^ (C=C) hybrid bond in the carbon ring and G peaks near 1580 cm^−1^ by the longitudinal vibration of the sp^2^ hybrid bond in the carbon chain or carbon ring by Gaussian fitting. These features are typical for amorphous carbon films [22].

Figure 3b shows the variations in the I_D_/I_G_ and G-peak positions of Samples 2–6. In general, I_D_/I_G_ ratio and G-peak position can characterize the DLC structure, which is related to the grain size and disordered structure [23]. The decrease in the I_D_/I_G_ ratio is an indicator of the decrease in the graphite-like phase concentration in the carbon films [24]. The G-peak displacement corresponds to the variation in the compressive stress of the film [25]; as the G-peak moves toward higher wavenumbers, the film compressive stress increases.

The figure reveals that the I_D_/I_G_ of the DLC film (Sample 2) with pure Si-B-C-N as the transition layer is much larger than that of the multilayer Si-B-C-N/DLC gradient films, indicating that the gradient films induce Si to form more Si-C bonds, with Si atoms connected with C atoms by a single bond to form the Si-C tetrahedral structure. In this case, the sp^2^ hybridization cluster in the film gradually decreases [26]. The Si phase has a larger atomic radius than other elements and is less likely to form π bonds, which can effectively inhibit the formation of aromatic ring structures and graphitization of the film. The figure further shows that the positions of the I_D_/I_G_ and G peaks of the multilayer gradient film first decrease and then increase as the number of layers increases. This also reflects, to some extent, the change in compressive stress within the film. The lowest I_D_/I_G_ (0.79) is found in the bilayer gradient films, which may be due to the suitable cluster size and tight stacking of the bombarded particles at the designated power change rate (1.5 W/min) and the increased number of contact sites, facilitating the formation of a denser structure and reducing the formation of defects.

### 3.3. Scanning Electron Microscopy (SEM) Analysis

Figure 4 shows the SEM morphology of the surface and cross-sections of each Si-B-C-N/DLC gradient film. Due to the relatively poor conductivity of multilayer Si-B-C-N/DLC gradient films, direct measurement would lead to the accumulation of a large amount of charge on the sample surface, which would affect the picture observation quality. This problem is solved by spraying gold on the sample surface to improve the conductivity of the sample and by connecting the sample to a metal base with conductive adhesive to conduct the charge from the sample surface. From the figure, there are no obvious defects on the surface of the film. The surface of the untreated film in Figure 4a is denser and more uniform than that of the other films; it is also smoother and flatter. The films shown in Figure 4c,e have the roughest surface. The surfaces of these films, which were obtained from the outermost Si-B-C-N target with a sputtering power of 130 W, show a number of islands of varying heights. This finding confirms that the size of the bombarded target particles gradually increases with increasing sputtering power, eventually leading to the deposition of film grains on the substrate. These larger particle clusters could be graphite clusters, nitride, and boride particles [14]. The cross-sections of the films are smooth and without obvious defects, and the demarcation line between the layers is not obvious owing to the gradient film preparation method applied. Sample 2’s film thickness is about 400 ± 50 nm. The total thickness of the gradient films is approximately 500 ± 50 nm, and their thickness increases slightly with the increasing number of layers. This may be because Sample 2 was prepared in a different way than the gradient film. The gradient film was sputtered simultaneously, while Sample 2 was sputtered sequentially. The difference in sputtering method and the rate of power change will affect the way the clusters are stacked and thus the thickness of the film. A columnar structure also appears, as shown in Figure 4e. This result indicates the spreading of the sputtered particle surface over a considerable distance and the epitaxial formation of individual grains into a uniform columnar crystal structure.

### 3.4. Optical Performance Analysis

DLC films have exceptional optical transmittance and do not affect the normal use of glass when used as a protective film for glass. A previous study showed [27] that films with higher sp^3^ bond contents have better optical transmittance. Figure 5 shows the UV-Vis transmittance spectra of the samples. The transmittance of each sample exceeds 65% in the visible range, and the highest optical transmittance of the single-layer gradient film reaches 97%. The figure also reveals that the optical transmittance decreases with the increasing number of gradient film layers.

In the Raman spectral analysis (Figure 3b), the bilayer gradient film shows the highest sp^3^ content, but its optical transmittance is less than that of the single-layer gradient film. This finding may be attributed to the size effect of clusters, which is another important factor affecting the optical properties of films. Because the microstructure and electron or mass density of the film vary with power, this directly affects the refractive index of the film and thus changes the optical transmittance of the film. In the present experimental protocol, as the power change rate increases, the internal cluster size of the gradient film also increases. An earlier study [28] found that this inter-cluster size effect has a considerable impact on the optical properties of films. Thus, as the number of gradient film layers increases, the transmittance of the film shows a decreasing trend.

### 3.5. Mechanical Property Analysis

Figure 6 and Figure 7 show the variation curves of the load, scratch depth, residual depth, acoustic signal versus scratch distance, and scratch morphology of Samples 1–6 with the same radius of curvature of the indenter. Here, the critical state of deformation of each sample is analyzed [29]. Table 2 details the *Lc* and final Rd of all samples. All samples in Figure 6 undergo plastic deformation at *Ly*. Samples 1 (Figure 6a and Figure 7a) and 6 (Figure 6f and Figure 7f) begin to break at *Lc*_2_, developing cracks at the edges of the scratches, and the film flakes off at *Lc*_3_ as the load increases. Sample 2 (Figure 6b and Figure 7b) shows slight film breakage at *Lc*_1_ during loading. As the load increases, striated cracks appear in the trajectory along the scratch, but no remarkable film flaking is observed. No significant fluctuations in the scratching depth, *Rd*, or acoustic signal profile of Samples 3 (Figure 6c and Figure 7c) and 4 (Figure 6d and Figure 7d) are found, and only traces of plastic deformation of varying degrees are observed on the films; no cracks, breaks, or peeling are noted. Sample 5 (Figure 6e and Figure 7e) shows a slight break after plastic deformation, and the cracks and breaks continue to increase at *Lc*_2_; however, the film remains attached to the substrate.

The above phenomena indicate that the gradient film design used to prepare the samples results in improved film–base bonding compared with that in pure DLC films, likely because a continuous change in structure occurs between the layers of the gradient film and the diffusion of elements between the layers forms a structure like a mechanical lock, resulting in a tighter and stronger bond between the layers [30]. Among the samples, the double-layer gradient film exhibits the best film–base bonding and only a slight increase compared with the single-layer film, as shown by the elastic recoveries obtained after scratching (Table 2; Rd = 50.32 and 43.06 nm). The XPS profiles of the samples indicate that the outer elemental content decreases as the sputtering power of the Si-B-C-N target decreases (Figure 2f), thereby reducing the elastic recovery ability of films. This trend in the binding force may be due to the slower rate of change in sputtering power for both targets (0.77 and 1.55 W/min) in the case of monolayers and bilayers, making the size difference between adjacent clusters smaller. After the formation of the double-layer gradient film, the bonding force begins to decrease with the increasing number of layers because the change rate of the sputtering power of the two targets increases (2.3 and 3.1 W/min). On the one hand, as discussed before in the Raman spectrum analysis in Section 3.2. the position of the G-peak shows a tendency to decrease and then increase, with the bilayer gradient films exhibiting the lowest position. This behavior is attributed to the element Si, which acts as an activator, reducing the internal stress and hindering the generation of the C-sp^2^ phase. On the other hand, according to the three-state model proposed by Ferrari and Robertson [31], a continuous change in sputtering power prevents the graphite structure inside the sputtered particles from undergoing sufficient structural transformation; thus, the hardness of the film decreases, and the film easily undergoes crack expansion when extruded by external forces, eventually leading to film peeling and cracking.

Friction and wear tests were conducted on each sample. The experiments were performed without added lubricant and using the same ambient temperature (300 K) at a humidity of 40% RH and 80% RH, respectively. The test results are shown in Figure 8A,B. Figure 8C,D shows the scratch condition of each sample after rubbing, with Samples 1, 2, 3, 4, 5, and 6 corresponding to (a), (b), (c), (d), (e) and (f) in the figure, respectively.

The experimental results in Figure 8A,B show that the coefficient of friction of the samples in the high-humidity environment is lower than that in the low-humidity environment. Meanwhile, the coefficient of friction of pure DLC films increased with time at both 40% RH and 80% RH humidity conditions. This is because over time, friction causes DLC film failure, and the increase in debris leads to a continuous increase in the coefficient of friction. The friction factors of Samples 2, 3, 4, and 5 are maintained at approximately 0.1 and 0.3, without any significant fluctuations. In contrast, Sample 6 showed greater fluctuations in both environments. The friction coefficient starts to decrease at 600 and 200 s and finally reaches a plateau because the surface of this sample is rougher (Figure 4e). Because the friction is not stable at the beginning of the test, owing to the interlocking effect between hard carbide [32,33], the curve shows larger fluctuations. As the sliding time increases, the bumps are gradually smoothed out and the friction factor stabilizes. In general, they decrease and then increase with the increasing number of layers. The bilayer gradient films exhibited a stable minimum value of 0.20 at 40% RH and a stable minimum value of 0.08 at 80% RH, whereas pure DLC film has a significant reduction.

The reason for the lower friction factor of each sample under a high-humidity environment (80% RH) than under a low-humidity environment (40% RH) is the water lubrication effect on sliding surfaces [34,35]. In addition, it has been shown that in a humid environment, water molecules can react with the DLC coating and form an oxygenated layer on the sliding surface. Water molecules will react with the DLC coating and form oxygen-containing hydrophilic groups on the sliding surface [36]. These hydrophilic groups will produce a water-rich surface layer that will provide lubrication for the docked surface. This result also indirectly verifies the previous study: films in environments with high humidity generally exhibit lower wear rates than in air environments [37].

From Figure 8C,D, under a low-humidity environment, the abrasive grains accumulate on the surface of the abrasion marks, and the grooves of the scratches become more obvious after local enlargement, accompanied by the micro-motion friction process, and resulting in abrasive wear. The high humidity facilitates the discharge of wear debris, and the amount of residual wear debris is very small, resulting in less contact between the sample and the debris. At this point, the wear mechanism of the Si-B-C-N/DLC gradient film is adhesive wear. Under adhesive wear conditions, the DLC graphitization process promotes the formation of a friction polymerization film, which has a nano-graphite cluster structure and adsorbs to the wear surface, exhibiting friction and wear reduction, thereby reducing micro-motion wear. It is noteworthy that white, snow-like cracks appeared on the surface of Sample 1 in both humidity environments. A long friction time leads to film flaking over a large area and substrate exposure. The failure form of this sample is surface fatigue wear. The film surfaces of Samples 2, 3, 4, 5, and 6 are relatively smooth, with no obvious failure characteristics. Slight grooves are found in the abrasion direction.

In summary, the friction coefficient of the gradient film is significantly lower, and the curve fluctuates smoothly compared with the pure DLC film, which is caused by the increase in hard carbide inside the gradient film and the formation of graphite slip layer on the surface due to the power change. The friction coefficients between the gradient films of each layer are similar, and the double-layer gradient film exhibits the lowest and most stable coefficient of friction under two different humidity environments. On the one hand, this is related to the higher bonding force of the film base; on the other hand, Raman spectroscopy reveals that the existence of compressive stress inside the bilayer gradient film also plays a buffering role, making the surface of the film less likely to be damaged. As a result, the unique gradient multilayer structure of the film has both a high bonding force and a relatively low coefficient of friction at this sputtering power conversion rate.

## 4. Conclusions

Multilayer Si-B-C-N/DLC gradient films were prepared on a glass surface and subjected to structural, optical, mechanical, and tribological property analyses. The conclusions of this research are as follows:The introduction of gradient films suppressed the formation of aromatic ring structures and promoted the formation of sp^3^ phases. As the number of layers increased, the sp^3^ hybrid bond content of the films initially increased and then decreased. Among the samples prepared, the bilayer Si-B-C-N/DLC gradient film showed the highest sp^3^ hybrid bond contents.The monolayer Si-B-C-N/DLC gradient film exhibited the highest optical transmittance (97%), which could be attributed to its sp^3^ content and the size effect of clusters within the film.Among the films prepared, the bilayer Si-B-C-N/DLC gradient films showed the highest film–base bonding (no film failure at a load of 200 mN) and low and stable friction coefficients (0.18 and 0.07) at two different humidity levels (40% and 80% RH).

This study mainly discussed the effect of sputtering power on the composition distribution and properties of gradient films. However, other process parameters also exert different degrees of influence on these properties. Further research should be conducted to examine the effects of parameters such as pressure, time, and temperature on the comprehensive performance of bilayer Si-B-C-N/DLC gradient films. This study provides a new approach to improve the binding force of glass-based DLC films and enriches the experimental data of DLC multilayer film systems.

## Figures and Tables

**Figure 1 materials-16-01665-f001:**
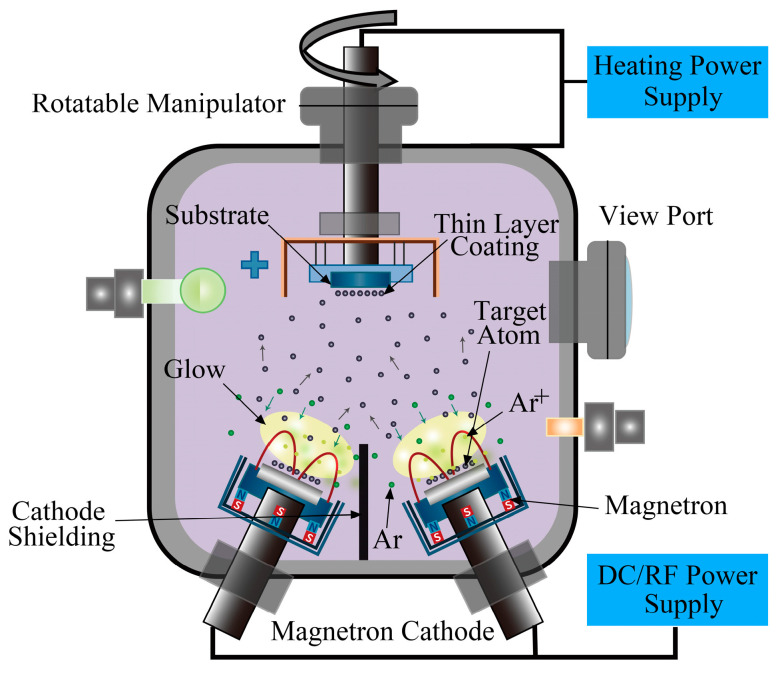
Schematic of the experimental setup.

**Figure 2 materials-16-01665-f002:**
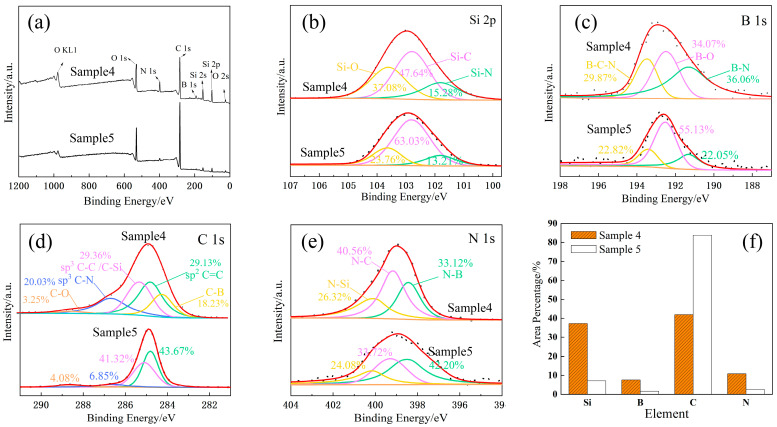
XPS spectra of Samples 4 and 5 and the content of each element. (**a**) Full Spectrum, (**b**) Si 2p narrow sweep, (**c**) B 1s narrow sweep, (**d**) C 1s narrow sweep, (**e**) N 1s narrow sweep, (**f**) Elemental content change.

**Figure 3 materials-16-01665-f003:**
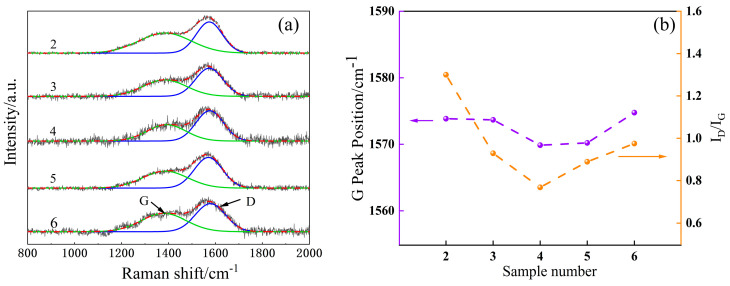
Raman spectra, I_D_/I_G_, and G peak positions of the Si-B-C-N/DLC gradient films. (**a**) Raman spectra of Samples 2–6, (**b**) Variations in the I_D_/I_G_ and G–peak positions of Samples 2–6.

**Figure 4 materials-16-01665-f004:**
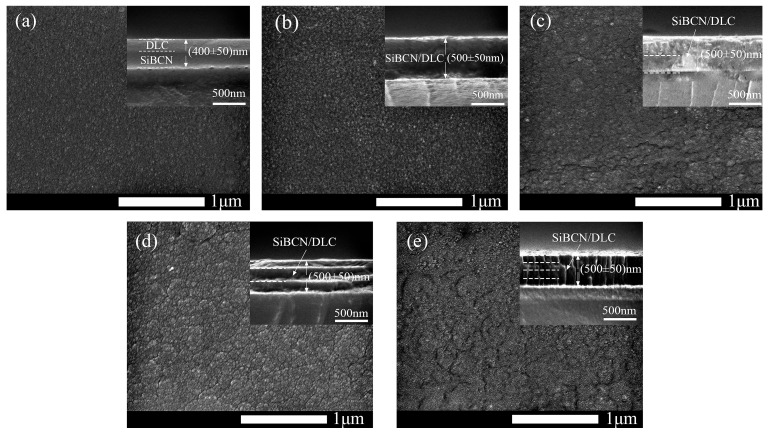
SEM micrographs of the Si-B-C-N/DLC gradient films. (**a**) Sample 2, (**b**) Sample 3, (**c**) Sample 4, (**d**) Sample 5, and (**e**) Sample 6.

**Figure 5 materials-16-01665-f005:**
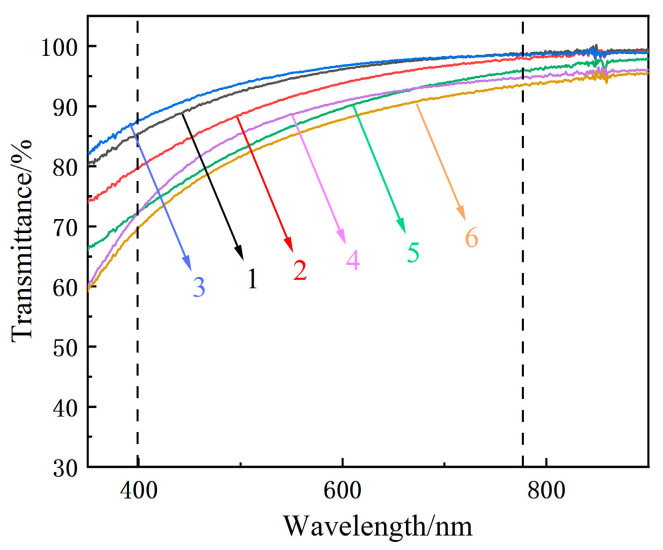
Optical transmittance of the Si-B-C-N/DLC gradient films; the visible wavelength range is within the two dashed lines.

**Figure 6 materials-16-01665-f006:**
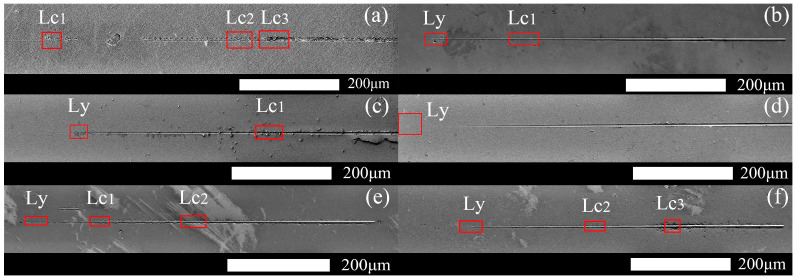
Residual scratch morphology of the Si-B-C-N/DLC gradient films. (**a**) Sample 1, (**b**) Sample 2, (**c**) Sample 3, (**d**) Sample 4, (**e**) Sample 5, (**f**) Sample 6.

**Figure 7 materials-16-01665-f007:**
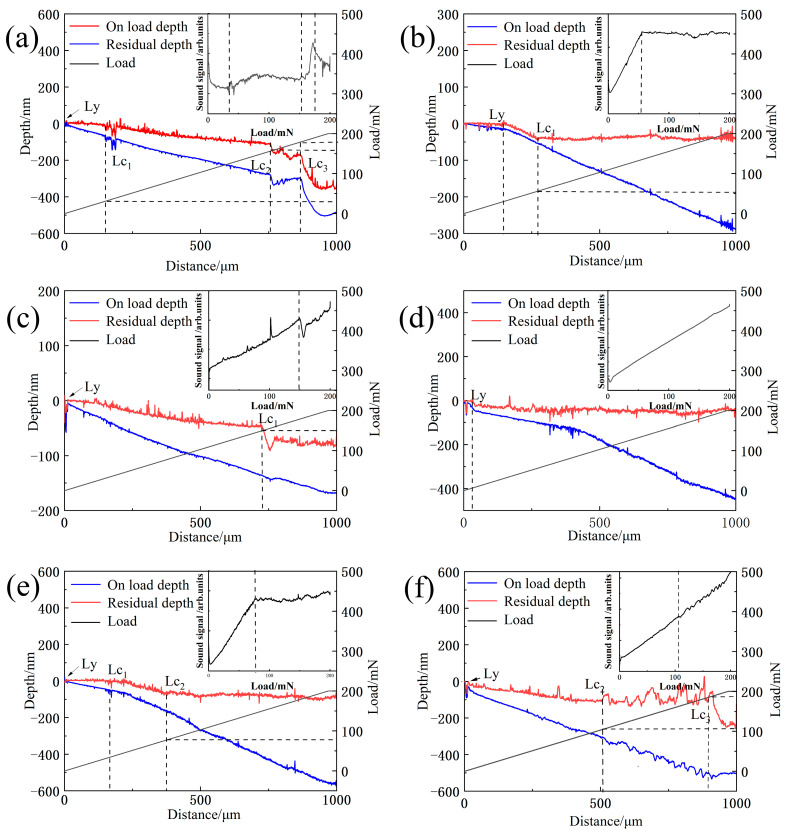
Variation curves of the load, scratch depth, residual depth, and acoustic signal versus scratch distance of the Si-B-C-N/DLC gradient films. (**a**) Sample 1, (**b**) Sample 2, (**c**) Sample 3, (**d**) Sample 4, (**e**) Sample 5, (**f**) Sample 6. *Lc*: critical load, *Ly*: plastic deformation of the film begins to occur, *Lc*_1_: a few cracks begin to sprout on the film surface, *Lc*_2_: scratch edges appear broken, *Lc*_3_: film is mostly peeled off, *Rd*: scratch depth. The dashed lines’ positions in the figure represent the corresponding scratch distance, load, and acoustic noise signals at the critical state.

**Figure 8 materials-16-01665-f008:**
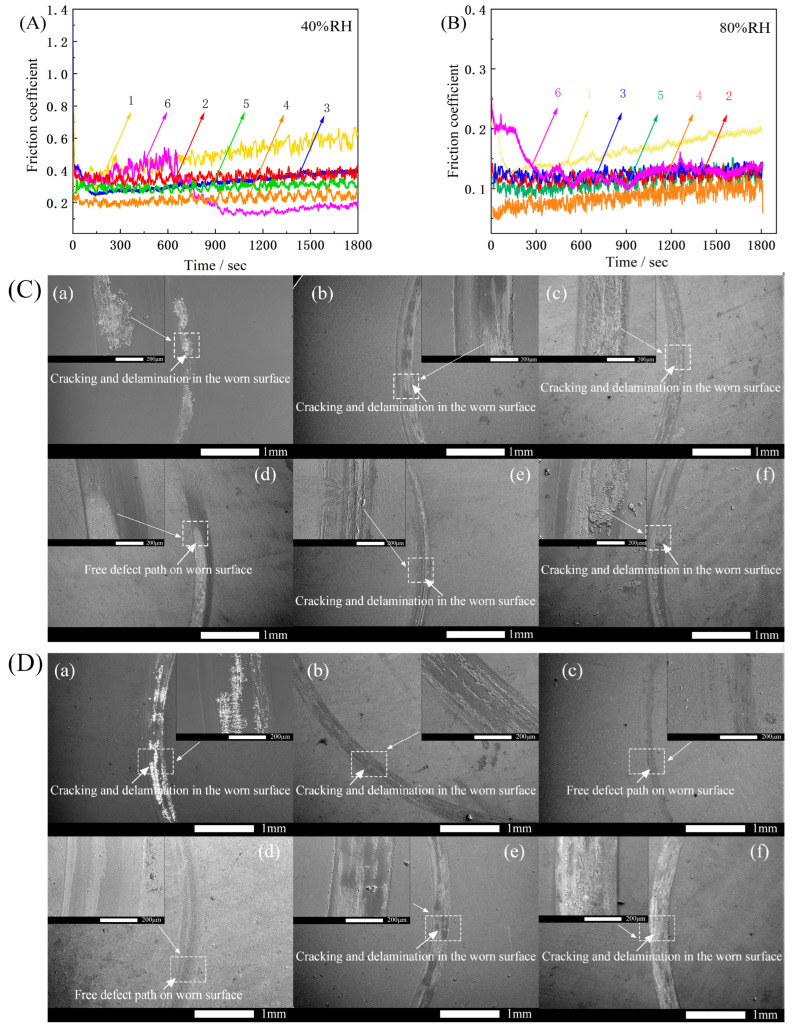
(**A**) Friction coefficient curves of Si-B-C-N/DLC gradient films at 40% RH. (**B**) Friction coefficient curves of Si-B-C-N/DLC gradient films at 80% RH. (**C**) Surface morphology of Si-B-C-N/DLC gradient film after wear at 40% RH. (**D**) Surface morphology of Si-B-C-N/DLC gradient film after wear at 80% RH.

**Table 1 materials-16-01665-t001:** Deposition parameters of the Si-B-C-N/DLC gradient films.

Sample Number	Deposition Parameters							Number of Gradient Film Layers	Sputtering Power Change Rate/W/min
	C Target		Si-B-C-N Target		Pressure/Pa	Argon Flow Rate/m^3^/s (×10^−7^)	Temperature/°C		
	Power/W	Time/Min	Power/W	Time/Min					
1	130	180	/	/	1.5	7	60	/	/
2	130	90	130	90	1.5	7	60	/	/
3	60 → 130	90	130 → 60	90	1.5	7	60	1	0.77
4	60 → 130 → 60	45 × 2	130 → 60 → 130	45 × 2	1.5	7	60	2	1.55
5	60 → 130 → 60 → 130	30 × 3	130 → 60 → 130 → 60	30 × 3	1.5	7	60	3	2.30
6	60 → 130 → 60 → 130 → 60	22.5 × 4	130 → 60 → 130 → 60 → 130	22.5 × 4	1.5	7	60	4	3.10

**Table 2 materials-16-01665-t002:** Critical load and residual depth of the Si-B-C-N/DLC films.

	*Lc*_1_/mN	*Lc*_2_/mN	*Lc*_3_/mN	*Rd*/nm
1	31.23	156.05	179.12	353.42
2	150.04	/	/	83.37
3	56.23	/	/	50.32
4	/	/	/	43.06
5	33.85	77.44	/	97.88
6	34.25	104.96	188.09	258.41

## Data Availability

Data openly available in a public repository.

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
