# Peer review of "Preparation and Performance of Multilayer Si-B-C-N/Diamond-like Carbon Gradient Films"

_materials, 2023, doi:10.3390/ma16041665_

Round 1

Reviewer 1 Report

This manuscript describes layered film of graded-composition layer on DLC. On the other hand, there are mistake and misleading in the measurements and its explanations, etc. Some technical problems are also contained in this manuscript. 

1. In Raman spectra, Explanation is wrong that variations in ID/IG correspond to variations in the sp2 and sp3 contents of a film. Visible Raman can detect only sp2 bonding because of its band gap.

2. There are some impurity peaks in the XPS survey spectra, but they are not assigned.

3. The X-ray source type on XPS should be indicated.

4. The type of glass substrate should be described in the specifications.

5. With films of these hardness, if a 500 gf load is applied to a 500 nm thick film, the substrate will have an effect to hardness value. For this reason, the obtained hardness values are not hardness for films.

6. Ambient conditions are important in sliding tests. There are no descriptions of these. In addition, there is no information on the condition of the sliding marks after sliding and the surface of the sliding counter parts. Thus, it's not clear what is worked to tribological properties.

7. The biggest problem in this manuscript is that the data are just listed. These should be organized to each other and make explicit what is novel. This manuscript does not contain any important and novel content as such an academic paper. All structural data should be considered holistically to discuss that what is influence to the properties. 

Some technical problems

8. “sp” which means bonding structure in carbon should be italic by rule of IUPAC. 

9. Variables should be written in italics. The characters which means a variable should be italic. This point is not only manuscript but also figures and tables.

10. In the case of parallel notation, the unit only with the last value is enough.

11. Units should be used by SI unit. “SCCM” or “gf” or some units are not Si unit. 

Author Response

Thanks to the professional opinions of the reviewers. According to your comments, I revised the article and answered the questions. The font color of the modified content is red in the article and this reply. Due to a large number of revisions in this paper, the original number of lines has changed, and the new revision position has been marked.

Reviewer 2 Report

The article is generally written in a clear and concise manner with a few typographical errors such as

·         Line 17 - 97% missing space at the unit

·         Line 214 - „( Film mostly…“ - extra space

·         Line 109 - for the diameter I recommend using “slashed O“ ∙

Comments:

1) Substrate transmittance - what thickness of substrate was used and for what wavelength was it evaluated (or is it the maximum transmittance value)? Because a maximum transmittance of 90% seems low to me for glass.

2) Sample preparation – Sample preparation data (Target-substrate distances, Deposition system geometry, Baseline pressure before deposition) need to be added

3) Hardness - There is now a trend towards reporting Vickers hardness in SI units (MPa or GPa) particularly in academic papers. Why are HV units used here?

4) Optical transmittance - Is the transmittance of the layers without substrate given here? Or why is the transmittance of the samples higher than that of the substrate? If YES is it necessary to add how it was calculated and that it is the transmittance of the layers themselves?

5) Mechanical testing - add exactly the type of ball and manufacturer

6) Figures 1,2,6,7 would deserve higher resolution (quality), figure 5 enlarged to full page

7) Figure 1 - For better interpretation of results it is necessary to add deconvolution also for sample 3. It would be useful to table the percentages of each type of bond.

8) Thickness - Why is sample 2 thinner (400 ± 50 nm) when it was prepared at 130 W from both magnetrons than samples 3-6 (500 ± 50 nm; other gradient 60-130 W) - see table 1. The time is for all 90 minutes (respectively 92 for sample 6).

9) Mechanical property analysis – Do we need to define Ly? Figure 5 - add sample labels to the caption or figure. The image is of poor quality, yet some of the damage (Lc) appears to have occurred earlier than indicated. Why are the whole scratches not shown.

10) There is misunderstanding in the paragraph with friction discussion (related to Figure 7). It is mentioned that samples 1-4 were measured, but then samples 5 and 6 values are mentioned. For a better discussion of the results the roughness parameters Sa and Sq measured e.g. by AFM would be useful. The reduction of the coefficient of friction in DLC is also due to the formation of a graphitic slip layer. Missing comparison/discussion with pure DLC (values here may be less than 0.03).

Author Response

Thanks to the professional opinions of the reviewers. According to your comments, I revised the article and answered the questions. The font color of the modified content is blue in the article and this reply. Due to a large number of revisions in this paper, the original number of lines has changed, and the new revision position has been marked.

Reviewer 3 Report

Duan et al. have presented the manuscript titled: Preparation and Performance of Multilayer Si-B-C-N/Diamond like Carbon Gradient Films. I suggest the authors to revise the manuscript according to the provided suggestions before being published. The suggestions are as follow;

1.      In the abstract section, please use the present tense rather past.

2.      Please add a sentence in the end of abstract, elaborating the practical amplification of this research work to make the manuscript more attractive for readers.

3.      As there are many articles available in literature regarding this material and specifically about the thin film of this material. In introduction section, authors should not just site those up to date articles but also elaborate their findings (result values) to compare their work with this report.

4.      Please mention the dimensions of the glass substrate in the method section.

5.      There exist a shifting in sample 4 towards higher wavelength as compared to sample 3 for C and N. Explain the reason. Second the X-axis and Y axis mentioned in Figure 1 is not clear, please amplify the axis.

6.      Similar to Figure 1, Figure 5 is also not clear, Please re-compile these figures and make them clear.

Author Response

Thanks to the professional opinions of the reviewers. According to your comments, I revised the article and answered the questions. The font color of the modified content is brown in the article and this reply. Due to a large number of revisions in this paper, the original number of lines has changed, and the new revision position has been marked.

Reviewer 4 Report

Title: Preparation and Performance of Multilayer Si-B-C-N/Diamond-like Carbon Gradient Films

Authors: Jiaqi Duan, Minghan Li, WenZhi Wang, Ziming Huang, Hong Jiang, and Yanping Ma

The manuscript entitled “Preparation and Performance of Multilayer Si-B-C-N/Diamond-like Carbon Gradient Films” by Jiaqi Duan et al. presents the fabrication of Si-B-C-N/diamond-like carbon (DLC) gradient films by radio frequency magnetron sputtering. Various experimental techniques have been used to analyze the structure and surface morphology of the samples. The manuscript could be recommended for publication after addressing all these suggestions and corrections.  

Q1: The introduction section should be modified with the advantages of gradient films compared to conventional films.

Q2: In table 1, the authors increase and decrease the deposition power from 60 W to 130 W to fabricate gradient films. I believe that by increasing and decreasing the deposition power, the microstructure and the electron density or mass density would differ throughout the films. Then it could change the refractive index of the film.

Q3: In the subsection, “XPS analysis,” the authors chose Sample 3 and Sample 4 for XPS measurements. The authors should compare Sample 4 and Sample 5 instead of Sample 3 and Sample 4. The thickness of the DLC films is different for samples 3 and 4. What is the incident photon energy for XPS? Remember, XPS is a surface-sensitive quantitative technique, and the information depth is a few nanometers. What is the thickness of the top DLC layers for Samples 3 and 4?

Q4: Authors should increase the font sizes of the text and labels for all figures (Fig.1, Fig. 2a, Fig.4, Fig.5, Fig. 6, and Fig.7) for better visualization.

Q5: On page no. 5 and line no. 176, The authors mention, “The films shown in Figs. 3c and 3e have the roughest surface”. It would be better if the authors analyzed the power spectral density profile of the surface SEM morphology and determined the surface roughness.  

Q6: In Figure 3, the authors show SEM images of the coated samples. I believe the resolution of the SEM measurement is ~10 nm, and the images look blurry. The authors should re-measure the SEM images for clarity.

Q7: The figure captions of Fig.5, Fig.6, and Fig.7 should modify with the detail about the sub-figures.

Q7: The acknowledgment should modify; authors should not acknowledge themself.

Author Response

Thanks to the professional opinions of the reviewers. According to your comments, I revised the article and answered the questions. The font color of the modified content is green in the article and this reply. Due to a large number of revisions in this paper, the original number of lines has changed, and the new revision position has been marked.

Round 2

Reviewer 1 Report

This manuscript describes layered film of graded-composition layer on DLC. On the other hand, there are still mistakes. Some technical problems are also contained in this manuscript.

About Last comment in #5

In particular, the indentation hardness test of thin films is determined by ISO14577, etc., and it is known that the stress of the indented tip propagates ten times as much. For this reason, the indentation depth should be less than 10% of the film thickness. Furthermore, all indentation depths are not same in this data. Indentation depth in samples 1 and 2 are in a lot because their hardness is a little low. For this reason, the state of stress propagation differs depending on the sample. Therefore, the crack analysis also does not make sense. In other words, if over 50% in depth indicated no meaning at all. These hardness data mislead the reader. I think it is necessary to delete the data or re-measure at 10% or less.

About Last comment in #6

For the sliding test conditions, the authors indicated 80% RH at 27°C. This is a highly humid condition. Normally, it is about 50% RH at 25°C. (ISO 554-1976) There are quite a few water molecules in the air. I think that several kinds of the elements included in this time are hydroxylated under this condition. (I can understand the oxygen peak in XPS if it was stored under these conditions.) Are these conditions really correct? please confirm. Also, if this condition is correct, make sure the effect of humidity. In particular, water has a large effect on the tribological properties of the surface.

Author Response

Thank the reviewers for their professional opinions. Based on your comments, I revised the article and answered the questions. The font color of the modified content is red in the article and this reply. Due to the large number of revisions in this article, the original number of rows has changed, and the new revision position has been marked.

Reviewer 3 Report

Authors have revised the manuscript very well according the suggestions provided by me. I would suggest now this article is ready to be published in the prestigious journal.

Author Response

Thanks to the professional opinions of the reviewers.

Reviewer 4 Report

The manuscript entitled “Preparation and Performance of Multilayer Si-B-C-N/Diamond-like Carbon Gradient Films” by Jiaqi Duan et al. presents the fabrication of Si-B-C-N/diamond-like carbon (DLC) gradient films by radio frequency magnetron sputtering and characterization using various experimental techniques. The manuscript could be recommended for publication in its present form.

Author Response

(The authors gave the same response as above.)
